# Sperm Competition and Paternity in the Endangered Firefly *Pyrocoelia pectoralis* (Coleoptera: Lampyridae: Lampyrinae)

**DOI:** 10.3390/insects15010066

**Published:** 2024-01-17

**Authors:** Xinhua Fu, Victor Benno Meyer-Rochow, Lesley Ballantyne, Xinlei Zhu, Qiyulu Zhang

**Affiliations:** 1College of Plant Science and Technology, Huazhong Agricultural University, Wuhan 430070, China; zhangqiyulu@163.com; 2Firefly Conservation Research Centre, Wuhan 430070, China; zhuxinlei32@gmail.com; 3Department of Ecology and Genetics, Oulu University, SF-90140 Oulu, Finland; meyrow@gmail.com; 4Agricultural Science and Technology Research Institute, Andong National University, Andong 36729, Republic of Korea; 5School of Agricultural, Environmental and Veterinary Sciences, Charles Sturt University, P.O. Box 588, Wagga Wagga 2678, Australia; lballantyne@csu.edu.au

**Keywords:** firefly, reproductive system, SSR, sperm competition, paternity, breeding, conservation

## Abstract

**Simple Summary:**

Breeding and re-introducing the endangered firefly *Pyrocoelia pectoralis* (Lampyridae: Lampyrinae) to a suitable habitat has become necessary because increased urbanization and light pollution have seriously reduced its numbers. To better understand this species’ reproductive biology, the anatomy of the male and female reproductive systems was investigated, and sperm competition and paternity were studied using SSR. The study of the anatomy of the female reproductive system as well as the paternity experiments revealed sperm competition in *P. pectoralis*. This is the first report to study sperm competition and paternity using SSR, both in fireflies generally, and in *Pyrocoelia pectoralis* (Olivier) in particular.

**Abstract:**

The endangered terrestrial firefly *Pyrocoelia pectoralis* (Olivier) is endemic to China. Populations of *P. pectoralis* have decreased dramatically due to urbanization and pollution. Breeding and re-introduction to a suitable habitat may save the species from becoming extinct. Because of its polyandrous character, an investigation into the possibility of sperm competition and paternity outcomes from multiple matings was initiated to better understand its reproductive physiology. To achieve these goals, 13 SSR markers were developed. The results of paternity experiments indicate there is a significant difference between P3 and P1 or P2. The female reproductive system has three spermathecae which accept sperm from different matings, and no bursa or spermatophore-digesting organ is developed. Our research established that multiple inseminations with sperm from different males occur, leading to competition between ejaculates. The benefits of such competition include an increasing number of sperm in the ejaculates of competing males and the consequential increase in fertilized eggs (thus, fecundity), and thereby a higher chance of genetic diversity and fitness in the offspring of the firefly *P. pectoralis*.

## 1. Introduction

Fireflies (Coleoptera: Lampyridae) are the most common representatives of terrestrial bioluminescent organisms [1,2]. The fascinating flash behavior in fireflies has attracted attention since antiquity. However, because of increasing urbanization and pollution [3,4,5,6,7,8,9], the populations of many species of fireflies have been declining [10,11]. The firefly *Pyrocoelia pectoralis* (Olivier) (Lampyridae: Lampyrinae) has terrestrial larvae and is endemic to China, where its populations have decreased alarmingly; it was listed as a protected animal for the first time in June 2023 (http://www.forestry.gov.cn/c/www/gsgg/509640.jhtml, accessed on 30 June 2023). Adults are sexually dimorphic and flightless females are considerably larger than winged males. During courtship, stationary females glow, and are also presumed to emit a sex pheromone to attract flying, bioluminescent males [12]. Copulations last 40–90 min, and both sexes can mate multiple times each night [12].

Natural populations of *P. pectoralis* show intense competition among males for access to the females, with several males chasing females and sometimes successfully interrupting copulating pairs [12]. Because a female can potentially mate with several different males [13], sperm competition in this species is likely to occur. In *P. pectoralis*, there is no spermatophore transfer during matings, but multiple matings with different males increase female fecundity and the eggs’ hatching rate [13]. Spermatophore transfer can enhance female longevity and reproductive fitness [13], but fireflies like *P. pectoralis* are not able to produce spermatophores, and it is not known what level of nutrition, if any, the ejaculated sperm could provide, or what benefits the female could accrue from multiple matings.

The complex anatomy of the females’ reproductive tracts in many insects may reflect selective pressures on polyandrous females to influence the use of sperm from different males for fertilization [14,15]. Female insects may affect the outcome of sperm competition through the utilization of different types of sperm storage organs, namely spermathecae [16]. In the female insects, the spermatheca is responsible for receiving, maintaining, and releasing sperm to fertilize eggs. The number and morphology of spermathecae vary according to species [15]. The spermatheca provides an appropriate environment that ensures the long-term viability of sperm, and maintaining sperm viability for extended periods within the spermatheca is crucial for an insect’s reproductive success [17]. In fireflies, the anatomy of the female reproductive tract is incompletely known [14,18,19,20], and in *P. pectoralis*, it has not been studied in detail at all.

Postcopulatory sexual selection includes competition among male gametes for fertilizations, called sperm competition, as well as preferential sperm use by females, known as cryptic female choice [21]. There are few reports on postcopulatory sexual selection (sperm competition) in fireflies, except that of Demary and Lewis [22], who used PCR-based RAPD (randomly amplified polymorphic DNA) genetic markers to assign paternity to offspring of doubly mated females of *Photinus greeni*. Studies of sperm competition require morphological or molecular markers that can be used to trace the paternity of several mating males. The absence of reliable, convenient and low-cost markers has slowed the research on sperm competition in fireflies.

To achieve this goal, genetic markers are needed, and microsatellites were selected as the markers of choice. Indeed, simple sequence repeats (SSRs) are characterized by high variability levels and relative ease of use at low cost [23]. Since no other nuclear genetic markers have been developed so far for the study of sperm competition or the genetic structure of *P. pectoralis*, SSR markers specifically for this firefly species have been developed in the present study to allow us to study the possibility of sperm competition and paternal success in *P. pectoralis*. Our research provides an insight into how female fireflies might influence the sperm competition from different males through the anatomy of the female reproductive tract, which is newly described here for *P. pectoralis*. An understanding of the intricacies of the reproductive physiology should help us to determine breeding strategies and thus enhance programs that address the reintroduction of endangered firefly species such as *P. pectoralis.*

## 2. Materials and Methods

### 2.1. Sample Collection and Feeding Scheme

Males and females of *P. pectoralis* were lab-reared for two generations. The original population of *P. pectoralis* was collected in Ezhou city, Hubei Province, China, in October 2018. *P. pectoralis* larvae were bred in transparent plastic circular boxes (20 cm diameter × 6 cm high) and provided with crushed land snails (*Bradybaena ravida sieboldiana*) as prey [12]. Male and female pupae were separated on the basis of the size difference of the visible light organs and elytral development (with females having very small elytra), and housed with moist filter paper in the same transparent plastic circular boxes mentioned above until emergence.

### 2.2. Anatomy of the Reproductive System

Freshly bred adults were frozen at −20 °C in 75% EtOH and stored until dissection [20]. Reproductive tracts removed from males and females were observed with a SZX 16 stereomicroscope (Olympus, Tokyo, Japan), and fat bodies attached to reproductive structures were carefully removed either by a fine brush or by soaking in cold 5% KOH for 1 min. For dissection of the male reproductive system, fine scissors were used to cut the abdomen laterally, and the intact male genitalia were removed to a transparent Petri dish for further dissection. Complete tracts were photographed with a DP72 CCD camera (Olympus, Tokyo, Japan). The female tract was examined carefully to determine the number of spermathecae and their characteristics. To determine the distribution of sperm in the differing spermathecae, ten females were selected, and each female was allowed to mate up to three times. Mated female were immediately dissected in a drop of phosphate-buffered solution (PBS, pH 7.2) to remove the spermathecae before sperm could be released to fertilize eggs from them. The spermathecae were transferred to separate histological slides together with a drop of PBS. Each spermatheca was then cracked open by using a fine needle to release sperm, which was fixed in 2.5% glutaraldehyde buffered with 0.1 M sodium cacodylate for 5 min and washed in running water. After drying at room temperature, the slides were stained for 15 min with 0.2 mg/ml 4, 6-diamino-2-phenylindole (DAPI) in PBS, washed, and mounted with Vectrashield (Vector Laboratories, Burlingame, CA, USA). These slides were examined with an IX71 epifluorescence microscope (Olympus, Tokyo, Japan), equipped with a BP360-370 nm excitation filter [24]. The presence of sperm in the spermathecae was then determined.

### 2.3. Target SSRs and Design of Multiplex Primers for Genotyping

The complete Hi-C genomes for *P. pectoralis* are available on NCBI GenBank with accessions JAVRBK000000000 under BioProject PRJNA1014999, and GMATA [25] was used to identify SSRs, the latter containing a repeat unit of 3~5 nucleotides, considered the target SSR marker site. The repeat numbers of units were defined 6 to 21 times. Then, 44 SSR primers were designed using Primer3 version 1.1.4 and synthesized by Sangon (Shanghai, China). The primer pair names and sequences are shown in Appendix A. Each forward SSR primer has a universal M13 19 bp tail sequence added at the 5′ end (5′-CAC GAC GTT GTA AAA CGA C-3′), according to the Model 4300 DNA Analyzer microsatellite analysis manual of LI-COR 4300. 

Initial testing for amplification and polymorphism involved genomic DNA of 60 adult male *P. pectoralis* individuals. The total genomic DNA of the samples was extracted using the CTAB (Cetyltrimethyl ammonium bromide) method [26]. Polymerase chain reactions (PCRs) were performed in 10 μL volume containing 5.28 μL ddH_2_O, 1 μL 0.2 mM dNTP, 1 μL 10 × Taq Buffer, 0.8 μL 3.5 mM Mg^2+^, 0.2 μL BSA, 0.2 μL of each primer pair (1.0 pmol/µL), 0.32 μL 1.0 pmol/µL IRDye 800-labeled M13f-29 primer (Li-Cor Inc., Lincoln, NE, USA), 0.2 μLTaq (Vazyme) and 1 μL template DNA. Amplification reactions were performed in a T100 Thermal Cycler (Bio-rad, Hercules, CA, USA) in the following sequence: 94 °C for 5 min; then 8 cycles of 30 s at 94 °C, followed by 30 s at 60 °C, which was lowered by 1 °C in each cycle, and 30 s at 72 °C; 27 cycles of 30 s at 94 °C, 30 s at 52 °C, and 30 s at 72 °C; and a 5 min extension step at 72 °C. Samples were kept at 4 °C until analysis.

For SSR polymorphism evaluation, PCR products were resolved using 6.5% polyacrylamide gel. Genotypes were read with a 4300 DNA analyzer (Li-COR, Lincoln, NE, USA). The number of examined individuals (N), numbers of different alleles (Na), effective alleles (Ne), Shannon’s information index (I), observed heterozygosity (Ho), expected heterozygosity (He) [27], unbiased expected heterozygosity (uHe), and polymorphic information content (PIC), as well as a summary of chi-square tests for the Hardy–Weinberg equilibrium were calculated using Cervus 3.0.3 [28].

### 2.4. Paternity Assignment

Paternity was calculated by taking the second male as the reference (two males competing with one female), with an index defined as “P2” (the index is between 0 and 1: when P2 is 0, all members of the offspring are attributed to the first male; when it is equal to 1, they are all attributed to the second male; it is equal to 0.5 when there is a promiscuous use of the sperm of the two males, with equal probability of both having fertilized the eggs) [29,30]. “P3” was calculated by taking the third male as the reference (three males competing with one female). Virgin males and females were subjected to two different protocols. In the first treatment, a virgin female was allowed to mate with two different virgin males in a plastic circular box with moist filter paper under 25 °C and a 12:12 photoperiod. Once the first male had finished mating and had separated, it was removed from the box, and the second male was introduced to allow to mate with the female (n = 16). In the second treatment, a virgin female was allowed to mate with three different virgin males using the same procedure as mentioned above (n = 9). Body lengths, widths, and weights of adult *P. pectoralis* insects were determined (Appendix A). Mated females were transferred to the same circular boxes to lay eggs on filter paper, and the eggs were allowed to hatch to first-instar larvae only. Parents and their corresponding offspring were collected separately. All samples (the parents and each of the larval offspring) were frozen at −80 °C until DNA extraction was performed. The genomic DNA extraction, PCR detection and genotype read methods were the same as in the description of Section 2.3. The loci with Ho > 0.75 were used for the detection of combinations (SSR308, SSR313, SSR320, SSR323, SSR324, SSR405, SSR504). The genetic diversity parameters of all loci were calculated by Cervus (V3.0.3) [28]. The parameters of the simulated offspring were set as 10,000, with the genotyping error rate being 0.01. 

## 3. Results

### 3.1. Anatomy of Reproductive System

In male *P. pectoralis*, the paired testes are located at the anterior end of the abdomen (Figure 1a), each thickly covered by red fat body and connective tissue. The long thin tubular seminal ducts (about 4 mm long and 0.6 mm wide), connected with the testes, lead to the seminal vesicles. Males have one pair of short tapering accessory glands (about 1.5 mm long and 0.3 wide at their widest point) (Figure 1a). 

The female *P. pectoralis* has two ovaries with numerous ovarioles at different states of maturation, with lateral oviducts converging onto a single median oviduct that enters the vagina. The vagina ends at the area where the median oviduct joins it, and three spermathecae of two different sizes, bound together by a transparent wall, arise from the ventral surface of the end of the vagina (Figure 1b–e). Beside the spermathecae, a small dual-branched female accessory gland enters the vagina (Figure 1b,c). There is no area that could be attributed to the bursa, and no spermatophore-digesting organ is developed (Figure 1b,c). At the anterior end of the vagina, three structures are noticeable: two small spherical spermathecae, and a much larger pear-shaped spermatheca with a smaller diameter of 1.3 mm (Figure 1c–f). The two small spermathecae (0.4 mm in diameter) have long coiled spermathecal ducts which unite and empty via a single duct very close to the opening of the larger spermathecal duct (Figure 1c,e,f). All three spermathecae were surrounded by a thin wall and covered by fat body (fat body not depicted in Figure 1b). The spermatheca S1 is flat in a female only mated once, whereas S1 is full in females that have mated three times. Identically shaped long-tailed sperm were detected in all three spermathecae, even if a female had only mated once (Figure 1g–j). 

### 3.2. Identification and Characterization of SSR Motifs

In total, 39,890 SSRs were identified from the *P. pectoralis* genome (Appendix A). The identified SSR motifs included dinucleotide (41.04%), trinucleotide (46.96%), tetranucleotide (5.34%), pentanucleotide (4.90%), and other (hexa-, hepta-, octa-, nona-, and decanucleotide) (1.77%) repeats, as shown in Appendix A. In the dinucleotide SSR, TA/AT was the most abundant type (6290 repeats) with a content of 38.42% (16,370 of all dinucleotide repeats), the TG/CA content was 30.40% (4977 repeats), the TC/GA content was 25.75% (4215 repeats), and the GC/CG content accounted for only 5.42% (888 repeats) (Appendix A). ATT/AAT was the most abundant type of trinucleotide repeat (15,578 repeats), with a content of 83.19% (18,725 of all trinucleotide repeats), followed by AGA/TCT (6.32%, 1183 repeats) and ACT/AGT (2.49%, 466 repeats) (Appendix A). Among the tetranucleotide repeats, the most abundant type was TTTA/TAAA (55.84%, 1190 repeats). Furthermore, the longest SSR repeats length was 600 bp (pentanucleotide TAACC), the shortest was 10 bp (dinucleotide), and the average SSR repeats length was 40.8 bp (Appendix A).

Repeat frequencies of SSR motifs of *P. pectoralis* were analyzed. As shown in Appendix A, the highest frequency for motifs with 5 tandem repeats was 19,874 (49.82%, 39,890 of all motif repeats), followed by motifs with 6 tandem repeats (19.73%, 7874) and 7 tandem repeats (10.63%, 4242). Dinucleotide repeat frequencies were 5–269, trinucleotide repeat frequencies were 5–51, and repeat frequencies were 5–78, 5–120, and 5–55 for tetranucleotide, pentanucleotide, and hexanucleotide repeats, respectively.

### 3.3. SSR Marker Development

We designed efficient SSR primers for paternity identification between different males. We mainly selected SSR sites with a 3–5 bp tandem motif with a repetition frequency of 6–21 times as target SSR markers, which have more conservatism and stability. Forty-four SSR primers were designed by Primer3 (Appendix A). The total genomic DNA of 60 adult male *P. pectoralis* individuals was used as the template. 

The results of developed 13 SSR markers are summarized in Appendix A and Appendix A. Among the 60 adult individuals in this study, each number of alleles of SSR was as between 4 and 34, with an average of 13.692 ± 2.052. The levels of observed heterozygosity (Ho) ranged from 0.400 to 0.900, with an average of 0.718 ± 0.047. The levels of expected heterozygosity (He) ranged from 0.594 to 0.940, with an average of 0.862 ± 0.028.

Polymorphism information content, or power of information content (PIC), is an index of the relative ability of the SSR marker’s genetic variability. The higher the polymorphism of a marker’s genotype, the higher the PIC value [31]. Polymorphic markers are considered highly informative (PIC > 0.50), reasonably informative (0.50 > PIC > 0.25), and slightly informative (PIC < 0.25). The levels of PIC ranged from 0.516 to 0.937, with an average of 0.845 ± 0.119. 

Significant (*p* < 0.001) deviations in the Hardy–Weinberg equilibrium (HWE) [32] were detected in three SSR loci: SSR325, SSR406 and SSR503 (Appendix A). In general, the SSRs here implemented showed high levels of polymorphism.

### 3.4. Paternity Assignment

The results indicate that after mating with two males, P2 (the second male paternity) is 0.4618 ± 0.0824 (n = 16, 1346 offspring detected), There is no significant difference between the first and second male (*p* = 0.5176 by *t*-test). After mating with three males, P1 (the first male paternity) is 0.1459 ± 0.0677, P2 is 0.2530 ± 0.0986, and P3 (the third male paternity) is 0.6012 ± 0.1013 (n = 9, 656 offspring detected). There is a significant difference between P3 and P1 with P2 (*p* = 0.00425) (Figure 2, Appendix A). 

## 4. Discussion

The female reproductive system in *P. pectoralis* differed from that of certain Luciolinae fireflies [20], being without a bursa and bursa plates (which have been shown to anchor a spermatophore within the bursa and protrude into the digesting gland). *P. pectoralis* females do not receive spermatophores or have a spermatophore-digesting organ that typically receives the male spermatophore in some lampyrid species [19,33], and this is the first instance of a firefly species having three spermathecae. Multiple spermathecae in female *P. pectoralis* could store many spermatozoa, and such structures are consistent with the females’ multiple-mating behavior. Anatomical observations indicated that multiple spermathecae in the female *P. pectoralis* reproductive system could accept sperm from different matings and different males, but each spermatheca containing only sperm from different males is highly unlikely. 

Nuptial gift material (other than gametes) provided by males to females during courtship or mating, including captured prey, seminal fluid protein, spermatophores, or various male body parts, is intimately tied to both precopulatory and postcopulatory sexual selection [1,34,35]. The flightless female *P. pectoralis* may utilize seminal fluid protein as a nuptial gift, and therefore additional matings could increase the fecundity and hatching rate [13]. Females are polyandrous, live for about 10 days and start laying eggs within 12 h of insemination [13]. This rapid turnaround would enhance survivability during the vulnerable stage of the life cycle.

However, South et al. [19] postulated a correlated evolution between flightlessness in the female and lack of reliance on nutritive material from the spermatophore. In this scenario, females that receive a spermatophore during mating also have modifications to their reproductive systems (the spermatophore-digesting gland and bursa with plates) that enable the female to derive additional useful material from the ejaculate of the males. Many of these females in the Luciolinae family are capable of flight [36]. It was postulated that the very large females of the type seen here in *P. pectoralis* have spent sufficient time as a larva to have accumulated an adequate supply of nutrition during their life cycle, thus reducing the need for extra input from males. It may be significant for this argument to point out that such females can lay eggs very soon after mating, thus reducing the likelihood that insects in the very susceptible stage (the large and heavy female) might suffer death due to predation. 

In the dimorphic Luciolinae firefly *Emeia pseudosauteri*, the flightless female is monandrous, has a spermatophore-digesting organ, one spermatheca (and three pairs of accessory glands in the male), and only accepts a spermatophore from one male during mating [20]. The female does not begin laying eggs until about 48 h post copulation, this time being equivalent to the length of time it takes the spermatophore to be digested in the spermatophore digestion gland or spermatophore digestion organ (SDG or SDO). This longer period leaves the female more susceptible to predation, and these females are not known to have obvious defensive capabilities. It would appear that selection would favor the polyandrous condition wherein the female receives sperm directly and can commence egg laying soon after the final mating. However, there are only a few genera in the Luciolinae family, wherein flightless females are known [36]. Flightless females like those of the genus *Atyphella* Olliff are no larger than the males, but similar to *Pyrocoelia* in which the abdomen is large and heavy, and the sizes of the elytra and hind wings are often severely reduced [36].

*P. pectoralis* females are polyandrous, and both fecundity and egg hatching success increase with additional matings [13]. Paternity outcomes where P3 has a greater percentage of the offspring may have a simple physical explanation. We do not yet know if there is any segregation of sperm into different spermathecae, but by the time of the third mating, it is possible that injection of a large quantity of sperm will fill each of the spermathecae closer to the opening of the ducts, and thus this P3 sperm will be more likely to be successful in fertilizing the eggs. 

Although it was recently reported that fireflies are affected by environmental pollution, especially light pollution [37,38,39,40], few researchers have focused on ways to restore dwindling firefly populations. Breeding of the firefly *P. pectoralis* and its re-introduction to areas in which it is very rare or from which it has disappeared are necessary steps in the conservation of the species. A low hatching rate of eggs has been observed when female *P. pectoralis* insects are allowed to mate with only one male. Multiple mating with different males can increase female fecundity and the egg hatching rate [13], in which sperm competition may play a key role. Multiple inseminations with sperm from different males, as have been shown [41], can lead to a competition between ejaculates, and the benefits of such a competition include an increasing number of sperm in the ejaculates of competing males and a consequential increase in fertilized eggs (thus, fecundity); this means a higher chance of genetic diversity and fitness in offspring. Comprehending the intricacies of reproductive systems in fireflies and the effects of sperm competition and paternity will be helpful in guiding breeding procedures. Without adequate genetic mixing and sperm competition, the likelihood of reduced fertility and resistance to environmental changes is likely to increase, and will therefore make an already small population even less viable and resistant to change.

## Figures and Tables

**Figure 1 insects-15-00066-f001:**
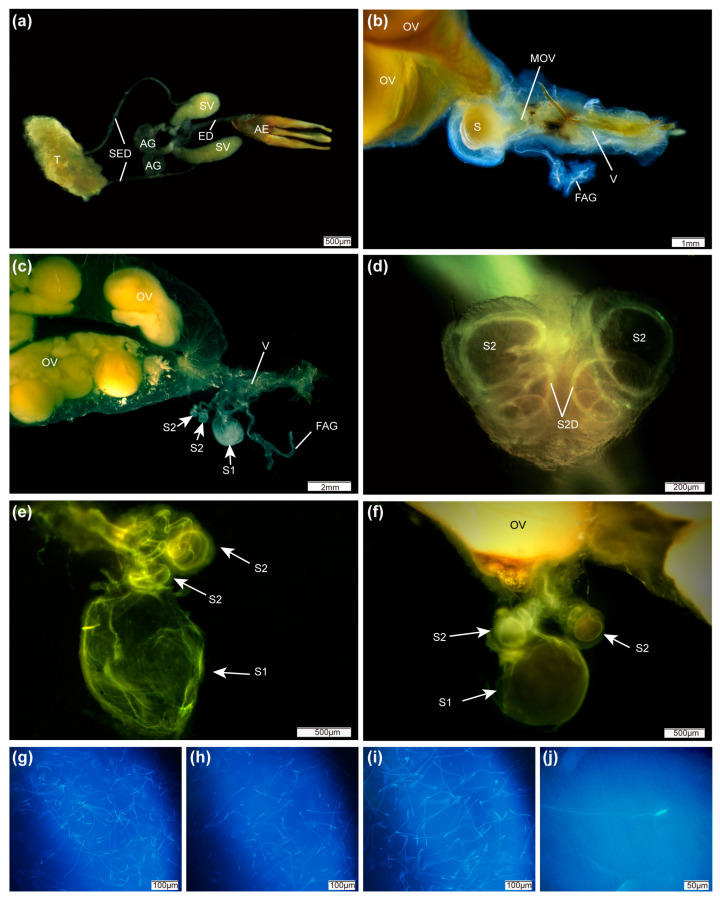
Anatomy of reproductive systems in *P. pectoralis*: (**a**) male reproductive system. T, testes; SED, seminal duct; ED, ejaculatory duct; SV, seminal vesicle; AG, accessory glands; AE, aedeagus; (**b**,**c**) female reproductive system (scale bar, 1 mm). OV, ovary; LOV, lateral oviduct; MOV, median oviduct; FAG, female accessory glands; S, spermatheca; V, vagina. (**d**) Two small (S2) spermathecae (scale bar, 1 mm); (**e**) structure and shape of spermathecae in female mated only once; (**f**) structure and shape of spermathecae in female mated three times; (**g**) sperm cell in S1; (**h**,**i**) sperm cell in S2; (**j**) individual sperm cell with long tail. (scale bar: **a**,**e**,**f** = 500 μm; **b** = 1 mm; **c** = 2 mm; **d** = 200 μm; **g**,**h**,**i** = 100 μm; **j** = 50 μm).

**Figure 2 insects-15-00066-f002:**
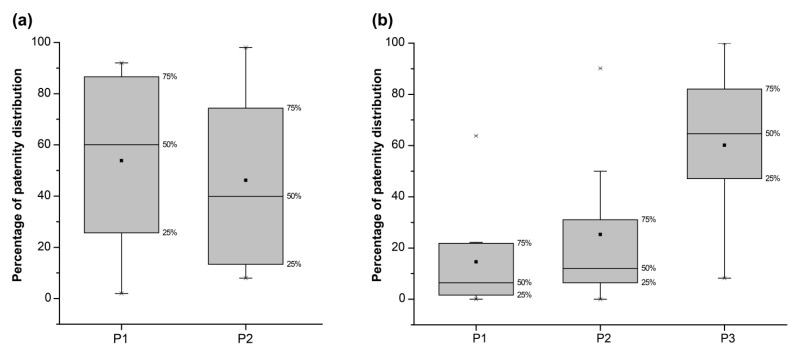
Box and whisker plot chart showing the percentage of paternity distribution at different times. The upper and lower edges of the boxes indicate the 75th and 25th percentiles, respectively. Whiskers represent the minimum and maximum, the line within the box indicates the median, the black spot within the box indicates the mean value, and the asterisks indicate the outliers. (**a**) Treatment 1: one female mated with two different males (P1 and P2); (**b**) Treatment 2: one female mated with three different males (P1, P2, P3).

## Data Availability

The datasets generated and/or analyzed during the current study are available in this published article and the Appendix A; the complete Hi-C genomes for *P. pectoralis* are available on NCBI GenBank, with accession number JAVRBK000000000 under BioProject PRJNA1014999.

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
