# Peer review of "Sperm Competition and Paternity in the Endangered Firefly Pyrocoelia pectoralis (Coleoptera: Lampyridae: Lampyrinae)"

_insects, 2024, doi:10.3390/insects15010066_

Round 1
Reviewer 1 Report
Comments and Suggestions for Authors
Author Response
We revised our ms and supplementary file follow reviewer1's comment. See attched file.

Reviewer 2 Report
Comments and Suggestions for Authors
Author Response
We revised our ms and supplementary file follow reviewer 2's comment. See attached file.

Reviewer 3 Report
Comments and Suggestions for Authors
The Manuscript entitled “Understanding sperm competition and paternity in the endangered firefly Pyrocoelia pectoralis helps saving this species” by Fu and Co-Authors deals with the paternity assessment in a polyandrous, endangered species of Lampiridae, endemic to China. Understanding whether sperm competition occurs and its effect on paternal success could be helpful in efforts to breed and conserve the species.
The study, carried out with morphological and molecular genetics approaches, reports for the first time the presence of three spermathecae in a species differing from others of the same family for the lack of the spermatophore digesting gland. The genetic analysis of the offspring in females mated with two or three males indicated that sperm precedence occured in the last case. The mode of sperm use by different males in relation to the sperm transfer/storage in the three spermathecae was discussed.
The paper is well written and all its sections are accurately described.
However, the points listed below need to be addressed. Thus, the MS may be accepted for publication in "Insects" after a minor revision.
Comments:
“Materials and Methods” section
Lines 100-101 – …“Ten mated females were selected before they could lay eggs”…….
• Authors are invited to explain (to a wide reader community) why they dissected the spermathecae from female before oviposition.
Lines 109-110 – …“The morphology of sperm from the spermathecae was then studied.”
• This phrase should be deleted because the study of the sperm cells is not supported by experimental data. Although the Figs S1 c, d, e, are representative for the presence of sperm cells (by nucleous staining) in the three spermathecae, they do not account for the morphology of the spermatozoa, which needs high resolution microscope techniques as SEM and TEM. However, it should have been possible to give dimensions and shape of the nucleus and the tail by light microscopy or, better, Nomarski optics, in addition to fluorescence microscopy. Authors, indeed, in the Figure S1f, merely show a single sperm cell in which the nucleous, after DAPI staining, appears fluorescent, with a trace of fluorescence resembling the tail (f). Moreover, the morphology of the sperm cells has not been discussed.
Line 123 - ….”CTAB method”…
· Insert the reference
“Results” section
Lines 163-164 – “Just anterior to this point a blind-ending bursa copulatrix without a spermatophore-digesting organ or bursa plates is developed (Figure 1c)”…
• The bursa copulatrix is neither indicated in the cited Figure 1c nor in figure S1 a-b. Authors are strongly invited to add this important information in the pictures.
Lines 164-172 and Figures 1-S1 - Whether Authors describe the shape of the two paired, small spermathecae, they do not assign a shape to the larger spermatheca (line 165, legend of Figure S1). Then, they define it as spherical (line 169). As it can be observed in the Figures 1 and S1 (included the schematic drawing) the large spermatheca is sac-like or pear-like shaped (not certainly spherical!).
• Authors have to clarify the shape of the larger spermatheca and indicate the measure of its minor diameter.
Lines 164-172 and Figures 1-S1 - Dimensions described in the text fo the three spermathecae are coherent with the scale bar of the Figure 1d. On the other hand, I seem that the scale bar (1mm) of figure 1c cannot be identical to that one of Figure 1d (1mm), as the female reproductive system of Fig. 1c is clearly at a lower magnification with respect to that of Figure 1d.
• Authors have to check the scale bar and correct it accordingly
Lines 164-172 and Figures 1-S1 - What is the “transparent membrane” (line 167) of the two small spermathecae? Is it the same ”membrane” defined “thin membrane” for all the three spermathecae (line 170)? Why only the the outer layer of the three spermathecae has been described and not their “core”, appearing whitish in the larger spermatheca or transparent (Figure 1d) or brown (Figure S1a) in the small spermathecae?
• Authors have to clarify these points. It would be better to use the generic term “wall” instead of “membrane”.
Lines 170-171 – …. “All three spermathecae were surrounded by a thin membrane and covered by the fat body (Figure 1c)”... The fat body covering the three spermathecae is not visible in the cited Figure 1c, as well as only one spermatheca is observable.
· Whether the fat body has been removed before capturing the image, indicate this structure as “not shown”.
Line 180 – Which spermatheca is indicated by the letter (S) in Figure 1c? The curved arrow from the spermatheca of Figure 1c towards Figure 1d is not explicative enough, as one spermatheca seems to be resolved into three spermathecae, accessory glands, common oviduct (and part of empty ovaries).
• Authors have to specify the type of spematheca visible in the picture. I also suggest to delete the curved arrow and better explain the legend of the Figures 1c and 1d.
Supplementary Materials
Line 306 and legend of figure S1 …”sperms” ……..”sperm”
• correct as sperm cells …. sperm cell
Typing errors
Line 51, spermtophore
• Correct as spermatophore
Line 103 Thespermathecae
• Correct as The spermathecae
Author Response
We revised our ms and supplementary file follow reviewer 3's comment. See attached file.

Round 2
Reviewer 1 Report
Comments and Suggestions for Authors
Reviewer’s comments to authors ver 2
insects
Manuscript No. 2765266
Title: 
Sperm Competition and Paternity in the Endangered Firefly Pyrocoelia pectoralis (Coleoptera: Lampyridae: Lampyrinae).
This time, English and sentences are much improved, but the section of “References” is confusing. After minor revision, this manuscript should be accepted for publication in “insects”.
Minor revision
1. Line 41; Leave a space between “alarmingly” and “and”.
2. Line 46; Delete “s” after “Copulation”.
3. Line 94; Add “,China” after province.
4. Line 156; Three males? Is the word (two males) correct?
5. Line 178; Move “about 4 mm long and 0.6 mm wide) before connected, because these explanations indicate the size of seminal ducts.
6. Line 186; Leave a space in front of “There”
7. Line 226; Leave a space between “were” and “5-78”.
8. Line 245; Comma (after SSR325) font is strange.
9. Line 267; Change to “bursa and bursa plates”.
10. Line 321; Adding “that” after “areas” makes the sentence easier to understand.
11. Line 407; GA P is correct? Anyway, write the number at the beginning of sentence.
12. Line 411; Write the number at the beginning of sentence.
13. Line 413; Write the number at the beginning of sentence.
14. Line 419; The 22nd paper (Demary and Lewis) duplicates with Line 411.
15. Line 421; This paper duplicates with Line 413.
16. Line 433; This paper duplicates with Line 407.
17. Line 435; This paper duplicates with Line 409.
18. Line 443; This paper duplicates with Line 405.
19. Note that the paper numbers vary widely.
Comments on the Quality of English Language.
Author Response
We revised our ms follow reviewer's comment. See attached file.

Reviewer 2 Report
Comments and Suggestions for Authors
Figure 1. Remove 'Mating behavior'.
Author Response
We revised our ms follow reviewer's comment. See attahed file.
